# The Performance of Concrete Made with Secondary Products—Recycled Coarse Aggregates, Recycled Cement Mortar, and Fly Ash–Slag Mix

**DOI:** 10.3390/ma15041438

**Published:** 2022-02-15

**Authors:** Katarzyna Kalinowska-Wichrowska, Edyta Pawluczuk, Michał Bołtryk, Jose Ramón Jimenez, Jose Maria Fernandez-Rodriguez, David Suescum Morales

**Affiliations:** 1Faculty of Civil Engineering and Environmental Sciences, Bialystok University of Technology, Wiejska 45 E, 15-351 Bialystok, Poland; e.pawluczuk@pb.edu.pl (E.P.); m.boltryk@pb.edu.pl (M.B.); 2Construction Engineering Area, School of Engineering Sciences of Belmez, University of Córdoba, 14014 Córdoba, Spain; p02sumod@uco.es; 3Inorganic Chemistry Area, School of Engineering Sciences of Belmez, University of Córdoba, 14014 Córdoba, Spain

**Keywords:** recycling technology, mechanical treatment, high quality recycled aggregate, recycled cement mortar, fly ash–slag mix, X-ray diffraction, thermal analysis

## Abstract

The properties of cement concrete using waste materials—namely, recycled cement mortar, fly ash–slag, and recycled concrete aggregate—are presented. A treatment process for waste materials is proposed. Two research experiments were conducted. In the first, concretes were made with fly ash–slag mix (FAS) and recycled cement mortar (RCM) as additions. The most favorable content of the concrete additive in the form of RCM and FAS was determined experimentally, and their influence on the physical and mechanical properties of concrete was established. For this purpose, 10 test series were carried out according to the experimental plan. In the second study, concretes containing FAS–RCM and recycled concrete aggregate (RCA) as a 30% replacement of natural aggregate (NA) were prepared. The compressive strength, frost resistance, water absorption, volume density, thermal conductivity, and microstructure were researched. The test results show that the addition of FAS–RCM and RCA can produce composites with better physical and mechanical properties compared with concrete made only of natural raw materials and cement. The detailed results show that FAS–RCM can be a valuable substitute for cement and RCA as a replacement for natural aggregates. Compared with traditional cement concretes, concretes made of FAS, RCM, and RCA are characterized by a higher compressive strength: 7% higher in the case of 30% replacement of NA by RCA with the additional use of the innovative FAS–RCM additive as 30% of the cement mass.

## 1. Introduction

The production of concrete in the world increases every year, which affects the high consumption of water, natural aggregates, and the production of cement. The cement industry alone accounts for approximately 4.1% of the EU’s and ~8 to 10% of the world’s anthropogenic CO_2_ emissions [1].

In 2017, about 45 billion tons of natural aggregates were produced, and it is estimated that the extraction rate will rise to 66 billion tons in 2025 [2]. It is estimated that the construction industry is directly responsible for the consumption of approximately 40% of the planet’s available resources, and of these, one-third of the material consumed is aggregates used by the cement product industries [3]. The consumption of aggregates is about 40 billion tons/year [4]. According to Eurostat data, the total amount of waste generated in Europe by households and companies by economic activity, according to the statistical nomenclature of economic activities of the European Community, is approximately 2535 billion tons per year, of which 36% (923 billion tons) is industrial waste from the construction industry [5]. More than one-third of all waste generated in the EU comes from the construction industry, of which up to 90% reaching landfills could be recycled and/or reused [6,7], and around 40–67% of construction and demolition waste is concrete [8].

Concrete production requires increasing amounts of raw materials, while industrial by-products can be used to meet this demand [9,10,11,12], either as supplementary cementitious materials or as alternative aggregates, as already proven by several researchers [13,14,15,16,17]. It could be beneficial for the environment to replace part of the cement clinker with supplementary cementitious materials, such as blast furnace slag [18], fly ash [19], and natural pozzolan [20] to reduce CO_2_ emissions [21]. Industrial waste from energy production is a huge part of the waste generated in the world’s industries. Fly ash and slag from the combustion of conventional fuels (hard and brown coal) are used to produce composite materials with a cement matrix, such as cement mortars or concretes [22,23,24,25,26].

The global production of coal fly ash is estimated to be 750 million tons per annum. Japan is the leader in the utilization of coal combustion by-products (utilization rate above 90%) [4]. The global average utilization rate of coal fly ash is about 25%, but in the European Union and the United States, it is significantly higher and has been growing for years [5,6]. For instance, in 2018, only 1.6% of coal fly ash produced in Poland was discarded [27,28].

In light of the data above, the use of waste materials in new concrete as a substitute for natural aggregates or cement is perfectly justified. Therefore, many authors have devoted their research efforts towards the use of recycling aggregates in new concretes, determining, for example, the most advantageous method of obtaining them [29,30,31,32]. Depending on the degree of purification of the aggregates from the recycling mortar compared with traditional concrete, the following results have been obtained: higher porosity and the increased water permeability of concrete with an increase in the proportion of RCA, or a decrease in compressive strength by up to a half [32,33,34]. On the contrary, others have found no deterioration of the strength parameters [35] or have even found an improvement [29,36]. Kazmi et al. [37] show that about a 32% decrease in peak stress of RAC than conventional concrete was reported. Additionally, a modulus of elasticity is around 40% lower than in concretes natural aggregates [38,39]. In turn, Munir et al. [40] in their modern research prove that confinement through steel spirals can significantly improve the performance of recycled aggregate concrete (RAC). In this study, the stress–strain behavior of steel spiral confined RAC incorporating acetic acid immersed and mechanically rubbed recycled coarse aggregates (AMRCA) was studied. Results show that the increase in confinement level increases the peak stress, peak strain, and ultimate strain of RAC incorporating AMRCA (AMRAC). Recycled cement mortar (RCM) produced in the recycling process is often used for research, e.g., as a substitute for part of the sand [41], and the result has proved that 15% or even up to 30% [42] of the required natural sand can be replaced by the recycled aggregate, mainly for the production of Portland clinker. An increase in compressive strength, a simultaneous decrease in the modulus of elasticity, and an increase in water absorbability in comparison with traditional cement mortars were observed. For years, research around the world has covered the use of fly ash and blast furnace slag in cement composites, mainly as a hydraulic additive [43,44] but also in geopolymer concrete [45,46,47,48], where pozzolana is indispensable for the polymerization process.

Fly ash from biomass has also become the subject of research for many scientists [27]. Biomass fly ash can be used in cement composite production after appropriate activation of the material. A study was conducted to assess the usefulness of mechanical and physical activation methods (grinding and sieving), as well as activation through the addition of active silica in the form of silica fume, as potential methods of activating biomass fly ash. The authors of [26] analyzed the possibility of using fly ash from the combustion of wood–sunflower biomass in a fluidized bed boiler as an additive in concrete. The results showed that fly ash applied in an amount of 10–30% could be added as a sand substitute in the production of concrete, without reducing quality (compression strength and low-temperature resistance) compared with the control concrete.

For the above-mentioned reasons, researchers are making efforts to study recycling opportunities in concrete and cement production. After all, recycling has three benefits: it reduces the demand for new resources, it cuts down on production energy costs, and it recycles waste that would otherwise be landfilled [7]. For instance, over the years, alternative binders such as blast furnace slag [49], fly ash [50], or silica fume [51], which are industrial by-products, have become valuable materials for concrete production. In addition to the benefits regarding waste production and the use of raw materials, CO_2_ emissions can also be lowered significantly, as less cement is needed for concrete production. To tackle the problems regarding the waste produced by the concrete industry, a great deal of research has been carried out regarding the use of recycled concrete rubble as a binder and an aggregate.

In line with the circular economy principle, huge amounts of concrete waste and the high availability of waste from the energy industry (e.g., fly ash–slag mixes), the solution to environmentally friendly sustainable development is the production of concrete with recycled coarse aggregates (RCA), recycled cement mortar (RCM), and fly ash–slag mixtures (FAS), which is explored in this article. The novelty of the article is to show the concrete properties with obtained secondary materials, received from one technological process and according to the comprehensive patented method [52].

## 2. Materials and Methods

### 2.1. The Process of Producing Recycled Aggregates (RA), Recycled Cement Mortar (RCM), and Fly–Ash Slag (FAS)

The preparation of cement composites, due to the two types of waste used (fly–ash slag and concrete rubble), was based on two main types of treatment: (1) the process of producing recycled coarse aggregate (RCA) and recycled cement mortar (RCM) according to the thermomechanical treatment of concrete rubble according to PAT.229887 [52] and (2) the mechanical treatment of FAS (drying and additional milling). The stages of the treatment processes of RCA, RCM, and FAS are shown below:

Stage I: Concrete rubble was crushed in a jaw crusher to a size < 40 mm.

Stage II: The rubble obtained in this way was placed in a thermal furnace and annealed with the following parameters according to PAT.229887 (650 °C for 60 min) [52].

Stage III: The debris, after being removed from the furnace, was placed in a Los Angeles drum and subjected to mechanical treatment according to PAT.229887 [52] to finally separate the cement mortar from the coarse grains of aggregate. The specific process was also described in [29].

Stage IV: The cooled material was sieved through a 4 mm sieve to separate the fine fraction (<4 mm) from the coarse fraction (≥4 mm). Coarse aggregate (≥4 mm) was additionally divided into 4–8 mm and 8–16 mm fractions ready for use in the new concrete mix.

Stage V: Recycled mortar (RCM) was milled again in the LA drum for 20 min.

Stage VI: Fly ash–slag (FAS) was milled in the LA drum for 20 min, after previously being dried at 60 °C.

The purpose of processing RCM and FAS in the ball mill was to increase the proportion of the finest particles in the material and thus increase their real surface area. RCM, which is thermally activated in the recycling process (PAT.229887) has pozzolanic properties, which were proven by [52].

This article presents the results of research on the physical and mechanical properties, and the microstructure of concretes prepared with FAS–RCM and RCA.

### 2.2. Materials

#### 2.2.1. Cement

Portland cement (CEM I 42.5R) meeting the requirements of the EN 197-1 standard “*Cement Standards—Part 1: Content, Requirements and Compliance Criteria of Commonly Used Cements*” was used in this study.

#### 2.2.2. Recycled Concrete Aggregate (RCA) and Natural Aggregate (NA)

The concrete rubble was obtained as a result of the thermomechanical treatment of pre-crushed concrete road curbs of the declared strength class C 30/37 received from a Polish industrial company. The rubble was sieved to obtain recycled aggregate separated into the 4–8 mm and 8–16 mm fractions, which were used in the following research. In Figure 1, the recycled coarse aggregates are shown.

In accordance with the research plan, the sand fraction of 0–2 mm, the gravel fraction of 2–4 mm, and dolomite fractions of 4–8 mm and 8–16 mm were used as the natural aggregates. The dolomite NAs are presented in Figure 2.

In Table 1, the properties of the NA and RCA aggregates are presented. RCAs were obtained as a result of thermomechanical treatment according to [29], as described in Section 2.1.

Figure 3 shows the gradation curves of the natural and recycled aggregates used for the concrete.

The gradation curves are similar because the recycled aggregate was divided into fractions and used in percentages, similar to the natural aggregate. Standard upper and lower curves are also shown.

The results above indicate that the recycled aggregate, obtained as a result of the thermal (650 °C) and mechanical treatment of concrete rubble, has properties similar to those of the natural aggregate. As a result of the treatment according to [29], the cement mortar that surrounded the aggregate grains was almost completely separated. Consequently, recycled aggregates with properties similar to those of the natural aggregates were obtained.

#### 2.2.3. Recycled Cement Mortar (RCM)

As a result of the thermomechanical treatment of the concrete rubble, as well as the RCA, RCM (<4 mm) was obtained.

To develop the higher specific area (as in the case of FAS) of the RCM, the additional process of milling in the LA drum for approximately 20 min was applied. The RCM after the additional milling process is ready to use in further tests.

Table 2 shows the sieving results of the recycled cement mortar (RCM) used for testing after the additional milling process.

#### 2.2.4. Fly Ash–Slag Mix (FAS)

The fly ash–slag mixture is post-production waste that was generated as an unavoidable by-product of energy production in conventional coal-fired power plants. The furnace waste was fly ash from electrostatic precipitators and slag from the slag removers, which, in the case of wet storage, forms ash and slag mixtures. The FAS obtained for the tests was characterized by high humidity (approximately 15%); therefore, before starting the tests, it was dried at 60 °C for approximately 24 h. After that, to activate the material (as in the case of the recycled mortar) and to increase the proportion of the finest fractions, the regrinding process was carried out in the LA drum for about 20 min.

In Table 3, the basic properties of CEM I 42.5 R, FAS, and RCM are presented.

#### 2.2.5. Superplasticizer (SP)

The Master Glenium SKY 638 superplasticizer based on third-generation polycarboxylate ethers was used for the tests. The correct configuration of the polymer chains allows the adsorption processes to be spread over time and avoids admixture build-up of cement hydration products. As a result, the fluidity effects of the concrete mixture are maintained without adversely affecting the early strength of the hardened concrete. The recommended dosage is 0.2–2.0% relative to the cement mass.

### 2.3. Methods

#### 2.3.1. Physical–Mechanical Tests of Concrete

The physical–mechanical tests of the concrete were conducted on 100 × 100 × 100 mm^3^ samples. The compressive strength test was carried out on six cubic samples following the EN 12390-3 standard “*Testing Hardened Concrete—Part 3: Compressive Strength of Test Samples*”. The tests of water absorption were performed according to Polish standard PN-88/B-06250 (Ordinary Concrete) on 6 and 12 samples (depending on the experiment).

The volume density in the dry and saturated state was tested according to EN 12390-7 standard “*Testing Hardened Concrete. Density of Hardened Concrete*”. The volume was calculated using the proven nominal dimensions of the molds, 100 × 100 × 100 mm^3^. The samples were weighed in a saturated, surface-dry state and dried to a constant weight after 28 days of maturation. The surface frost resistance test was carried out for Series 3, 9, and 10. This test consists of determining the mass of the exfoliated material from the surface of the sample covered with a 3% NaCl solution after a given number of freezing and thawing cycles. The test procedure according to PKN-CEN/TS 12390-9 corresponds to the description given in the standard for precast roading concrete (EN 1339:2003 “*Concrete Paving Flags Requirements and Test Methods*”). Thermal conductivity tests were prepared according to the EN 1946-3:2000 standard “*Thermal Performance of Building Products and Components—Specific Criteria for the Assessment of Laboratories Measuring Heat Transfer Properties—Part 3: Measurements by Heat Flow Meter Method*”.

#### 2.3.2. Physical–Chemical Characterization of the Raw Materials and Their Microstructural Properties

To determine the chemical composition of the raw materials (cement, natural aggregate, recycled concrete aggregate, recycled cement mortar, and the fly–ash slag mix), X-ray fluorescence (XRF) was performed. ZSX PMIMUS IV (Rigaku, Japan equipment was used. All the raw materials were characterized by X-ray diffraction (XRD) carried out with an instrument with CuKα (λ = 1.54050 A; 40 kV; 30 mA). Diffraction patterns were measured between 10° and 70° at a rate of 1 2σºmin. The morphology of two samples (with only natural aggregate or with RCA) were obtained using scanning electron microscopy (SEM) and backscattered electron (BSE) imaging. A JEOL JSM 7800F (JEOL, Mitaka, Japan) was used. Small portions of the samples were cut and then placed in epoxy resin for 48 h. After 48 h, the samples were polished and sputtered with carbon. Furthermore, simultaneous thermogravimetric analysis and differential thermal analysis TGA/DTA were carried out for three hardened samples. TGA/DTA was performed at a heating rate of 5 °C·min^−1^. The working temperature ranged from ambient temperature to approximately 1000 °C.

## 3. Experimental Design

The experiment was carried out in two stages. In the first stage (Experiment I), the most favorable content of the concrete additives in the form of RCM and FAS was determined experimentally, along with their influence on the physical and mechanical properties of concrete. For this purpose, 10 test series were carried out according to the experimental plan described in detail in Section 3.1. After determining the most favorable content of waste additives, in the second stage of the tests (Experiment II), three successive series of concrete were tested, where the recycled aggregate was also used alongside the best experimentally determined content of waste additive. Details and results of the second stage of the study are included in Section 3.2.

### 3.1. Experiment I: Determining the Most Advantageous Content and the Effect of the FAS–RCM Additive on the Properties of Concrete

To better determine the best amount and the impact of FAS–RCM on the concrete’s properties, the experiment was carried out based on two variables:

*X*_1_: the amount of RCM (0 ≤ *X*_1_ ≤ 50 as a percentage of the summary mass of FAS–RCM);

*X*_2_: the amount of FAS–RCM (0 ≤ *X*_2_ ≤ 30 as a percentage of cement mass).

The amounts of RCM and FAS resulted from the authors’ previous experiences and the limit of the content of additives as a cement substitute according to the EN-206 standard. The details of the range of variability and the levels of the analyzed factors are presented in Table 4. Laboratory tests were carried out according to a complete plan for 2 variable factors at 3 levels of variability. The starting quantities were, inter alia, concrete frost resistance determined based on the loss of concrete mass by peeling under the influence of 28 cycles of freezing and thawing, compressive strength after 28 days, water absorption, volumetric density, and the λ coefficient of concrete. Laboratory tests were used to obtain information on the impact of the FAS mixture and the mortar from recycling concrete rubble subjected to thermomechanical RCM treatment on the physical and mechanical properties of concrete.

Additionally, a control series was made without the addition of FAS–RCM. The test experimental plan is presented in Table 5.

Table 5 shows the percentage distribution of the variables for each series. In total, 10 different series of samples were prepared.

#### 3.1.1. Concrete Mix Composition

All recipes maintained a constant *w*/*s* ratio of 0.45 with the following constant aggregate proportions: sand constituted 35%, the 2–4 mm fraction made up 10%, the 4–8 mm fraction made up 25%, and the 8–16 mm fraction made up 30%. The variables were the content of recycled concrete mortar (0–50%) and the content of the FAS–RCM mixture (10–30% *w*/*w*).

The additives were introduced into the concrete in the same way as fly ash. It was assumed that the additives would be included in the minimum amount of cement in concrete and the equivalent water–cement coefficient would be determined by the coefficient k, the size of which depends on the class and type of cement. The maximum amount of additive (FAS–RCM) included in the k value was as follows: FAS–RCM/cement (C) ≤ 0.33 (by weight).

For the strength class of CEM 42.5 cement, the value of k = 0.4 was established. The minimum cement content required in the appropriate exposure class may be reduced by a maximum amount of k × (minimum cement content in a given exposure class—200) kg/m^3^.

As a result, 40% of the planned cement mass was replaced with FAS–RCM, and the aggregate made up the remaining part. The starting series for the concrete composition design was the control series (No. 10), which did not include additives. The initial amount of cement was assumed to be 360 kg/m^3^, meeting the requirements of all exposure classes according to EN 206:2014 and accounting for strong chemical aggression (the concrete was exposed to contact with soil and groundwater). We also assumed a *w/s* ratio of 0.45, which fell within the permissible range, and the minimum concrete class was C35/45. The final compositions of the concrete mixtures are presented in Table 6.

#### 3.1.2. Preparation of the Concrete Samples

The concrete mix for the samples in the tests was prepared in a laboratory counter-rotating concrete mixer. Each preparation was about 18 dm^3^. First, weighed aggregate, cement, and possibly FAS and RCM, were poured into the mixer drum and mixed until dry. The mixing water was dosed with the superplasticizer and mixed until the material had a homogeneous consistency. For the preparation of samples, in each of the test stages, cube-shaped forms with dimensions of 100 × 100 × 100 mm^3^ (Figure 4), meeting the requirements of EN 12390-1:2013, were used. The samples were made and cared for following the EN 12390-2:2011 standard. Compaction took place on a vibrating table with a vibration frequency of 50 Hz (6000 cycles per minute), an amplitude of 0.5 mm, and an acceleration of 5 g (g = 9.81 m/s^2^). The vibration time was 15–30 s.

After the samples had been stored, they were stored for a minimum of 16 h in molds covered with glass fleece in order to limit water loss from the surface of the samples. After demolding, the samples were stored in water at a temperature of 20 ± 2 °C until the tests were carried out.

### 3.2. Experiment II: Determining Properties of Concrete with FAS–RCM and RCA

In order to determine the impact of RCA with the addition of FAS–RCM on the strength properties of concretes, another experiment was prepared, which used RCA derived from waste generated in the production process of elements at Jadar Sp. z o. o. The basis of the concrete rubble was dolomite aggregate. The following series of concrete mixes were prepared:Series 1: made solely from sand and dolomite aggregate (control series).Series 2: 30% of the NA (>4 mm) was replaced with recycled aggregate based on dolomite. This is the acceptable percentage of replacement of coarse aggregate with recycled aggregate for exposure classes other than X0 permitted by the EN 206:2014 standard.Series 3: 30% of the NA (>4 mm) was replaced with recycled aggregate based on dolomite, and an additive was introduced in the form of a mix of FAS–RCM, where FAS and RCM each accounted for 50% of the weight of the additive.

Table 7 presents the test plan and the compositions of each concrete series for 1 m^3^ of the mixture.

## 4. Test Results and Discussion

### 4.1. Characterization of the Raw Materials

To know the chemical composition of the raw materials, XRF was performed (Table 8). In the cement used, the major compound was CaO from the cement clinker. A similar result was found by Suescum-Morales et al. [53]. For NA, the major compound was CaO. This indicates that the nature of the sand is limestone [54]. However, for the RCA, the SiO_2_ and CaO values were close, which may indicate that this RCA was obtained from siliceous aggregates. Very similar values were also obtained for RCM. For the fly ash–slag mix, the major compound was SiO_2_. This result is similar to that of Goncalves et al. [55].

Due to the variability in the composition of natural aggregates, recycled concrete aggregate, recycled masonry aggregate, and the fly ash and slag mixture, Figure 5 shows the XRD patterns of the raw materials used in this work. The main phase for the natural aggregate was dolomite (11-0078). The results also found quartz (05-0490) and calcite (05-0586) [56]. This result agrees with those found by XRF.

For recycled concrete aggregate, the phases found were quartz (05-0490), calcite (05-0586), portlandite (44-1481), illite (02-0050), and larnite (09-0351) [56]. The same phases were found by Suescum-Morales et al. [1] for recycled masonry aggregate. For recycled cement mortar, the same phases were found as for recycled concrete aggregate. This result is logical, as they are the same material, only sieved. The fly ash–slag mix was observed to be an amorphous compound. The main phases were quartz (05-0490) and mullite (15-0776). Magnetite (19-0629), hematite (33-0664), and anhydrite (37-1496) were also found. In addition, the phases detected for the cement were hatrurite (86-0402), brownmillerite (11-0124), calcite (05-0586), gypsum (21-0816), and portlandite (44-1481) [56], which were included. Similar results have been reported by several authors [45,56,57,58].

### 4.2. Experiment I: The Results of Concretes with the Addition of FAS and RCM

#### 4.2.1. Compressive Strength

The average test results of the compressive strength for each series after 28 days of curing are shown in Figure 6.

The changes in the concrete compressive strength depending on the amount of RCM (*x*_1_) and the amount of FAS–RCM (*x*_2_) are presented in Figure 7. The function describing the dependence of the compressive strength on the tested variables for concrete with FAS–RCM is expressed in the following equation (Equation (1)):(1)$$Y=64.203+2.432x_{1}+1.452x_{2}\;\;\;\;\;\;\;\;\;R^{2}=0.94$$

Figure 8 shows that the presence of additives in the FAS and RCM mixture had a positive effect on the compressive strength of concrete after 28 days of maturation. With an increase in the content of the FAS–RCM mixture from 10% to 30%, an increase of 5% in strength on average was observed after 28 days of maturation. On the other hand, increasing the content of recycled mortar in the FAS–RCM mass from 0 to 50% resulted in an increase in compressive strength by an average of 8–14%. It should also be noted that in the presence of the highest analyzed content of RCM (50% by weight of FAS–RCM) higher strengths were observed compared with the control series (No. 10). Series 9 containing 30% FAS–RCM mass and 50% recycled mortar showed an average increase in strength of 10% after 28 days compared with the control series. This confirmed the active properties of the mortar subjected to thermomechanical treatment, as well as the ash and slag mixture. Probably, increasing the specific surface area of these materials by grinding them for longer would enhance this beneficial effect.

#### 4.2.2. Volume Density

In Figure 8, a graphic interpretation of the results obtained by the concrete volume density test in a saturated and dry state is shown.

As can be seen in Figure 8, the differences in the dosage of the ash–slag mixture additive and the recycled concrete mortar did not significantly affect the bulk density of the composite. In individual series, differences in density at the level of 1–2% were observed, which is within the measurement error. The saturation of concrete increased the bulk density by approximately 4% compared with the dry density, which corresponds to the level of concrete absorbability. However, it was observed that in both cases, the highest densities were recorded for the control series, which resulted from the lower density of the additives used (FAS and RCM) compared with the cement’s density.

#### 4.2.3. Water Absorption

The average results of concrete water absorption tests are presented in Figure 9.

As shown in Figure 9, along with the increase in the content of the FAS–RCM mixture, a slight decrease in the water absorption of concrete was observed, which resulted from the sealing of fine aggregate fractions by the applied additives. At the highest content of FAS and in the presence of recycled mortar, a minimal increase in the absorbability of the composite was recorded, which was partly caused by the difficult compaction of the concrete mix. The presence of the additives used at the highest content had an impact on the workability of the concrete mix; therefore, the superplasticizer content or the w/c ratio should be increased in these series. The lowest water absorption was recorded for Series 9 containing 30% of the FAS–RCM blend and 50% RCM content in the additive.

Generally, however, it should be noted that the differences in water absorption throughout the experiment were small and amounted to a maximum of 0.9% and did not exceed the total level of 5%.

#### 4.2.4. Frost Resistance

Concrete’s freezing and thawing resistance was tested with the use of de-icing salts for three selected series (3, 9, and 10). The test results after 28 cycles of freezing and thawing are summarized in Table 9.

As shown by the test results (Table 9), in all series, the mass of exfoliation did not exceed 1 kg/m^2^. In the presence of 30% FAS (No. 3), the weight of the exfoliated material was reduced by 5% compared with the control series. On the other hand, the presence of 50% cement mortar in the FAS–RCM mixture resulted in a decrease in the mass of exfoliated material by an average of 13% compared with the control series. This proves the beneficial effect of the presence of the FAS–RCM mixture on sealing the concrete structure. All series of concretes can be classified as the same category of surface frost resistance as concrete in the FT1 pavement (average weight of exfoliation ≤ 1.0, no result > 1.5).

### 4.3. Experiment II: The Test Results of Concrete with FAS–RCM and RCA

#### 4.3.1. Compressive Strength

In Figure 10, a graphical interpretation of the results of the compressive strength test after 28 days of maturation is presented.

As shown in Figure 10, the replacement of 30% of the natural (dolomite) aggregate with recycled aggregate (Series 2) resulted in an increase in the compressive strength of concrete by approximately 7%. Due to the high degree of removal of the cement mortar from the surface of the recycled aggregate as a result of the applied thermomechanical treatment, a high-quality product with parameters comparable with crushed dolomite aggregate was obtained. This favorable tendency shows that recycled aggregate can replace even a large part (up to 100%) of natural aggregate; however, at the moment, this is not allowed by the PN-EN 206:2014 standard. The presence of a small amount of the remaining cement mortar improved the connection of the aggregate with the new cement matrix, which sealed the aggregate–slurry contact zone and thus increased the compressive strength of the concrete. The presence of larnite and portlandite found by XRD for RCM, as shown in Figure 5 (unset cement), improved the compressive strength. In Series 3 containing the recycled aggregate and the FAS–RCM mix as 30% of the cement mass, an increase of approximately 7% in compressive strength was recorded compared with the series containing natural aggregate. On the other hand, compared with Series 2 (where recycled aggregate was used), the increase in strength was small, about 2% (a difference within the error limits). However, this may constitute a premise for further research on having a greater proportion of the FAS–RCM mixture in concrete. Once more, in this case, the presence of portlandite and larnite improved the compressive strength. As it has been proven by authors in previous experiments in the article [29], the coarse aggregates from the comprehensive recycling process obtained according to the patented method [45] (after thermomechanical treatment) are high quality and improve the strength parameters of new concretes.

#### 4.3.2. Water Absorption

In Figure 11, the results of the mass water absorption of concrete samples are presented for the control series, the series with recycled dolomite aggregate, and the series using recycled aggregates and the FAS–RCM additive.

As shown by the results of the tests conducted (Figure 11), the water absorption of all analyzed concrete samples did not exceed 5%. Moreover, the highest water absorption value was recorded for the control concrete, made without recycled substrates (3.67%), and the lowest was found for concretes with the use of recycled aggregates, FAS–RCM (3.46%). It should be noted that the water absorption of concretes made with recycled aggregates was only slightly higher (less than 1% difference). It can therefore be concluded that high-quality recycled aggregate cleaned of recycled mortar fits perfectly into the cement matrix and that the addition of FAS–RCM additionally seals the structure of the composite, because of which the absorbability of concrete is much lower than that allowed by the standard.

#### 4.3.3. Volume Density in Dry and Saturated States

The results of the bulk density in the saturated and dry states are presented in Figure 12.

As can be seen from Figure 12, the concrete density values of the individual series were similar. Series 2 and 3 concretes, made with recycled aggregates and recycled additives (FAS–RCM), were characterized by the lowest density. Compared with the control series, there was a ~2% decrease in both saturated and dry concretes. In individual series, differences of 1% in the density were observed, which is within the measurement error. The saturation of concrete increased the bulk density by approximately 3–4% compared with the dry density, which translates into the percentage of concrete absorbability obtained. The highest densities were recorded for the control series, which resulted from the lower density of the additives used (FAS–RCM) compared with the density of the cement. The use of a larger amount of recycled aggregates due to their high quality (even up to 100%) and the addition of FAS–RCM may result in a further reduction in concrete density, and consequently, a lower specific weight of the elements could be produced.

#### 4.3.4. Thermal Conductivity Index

Figure 13 shows a graphic interpretation of the results of the thermal conductivity coefficient test of the material tested under dry conditions.

Figure 13 shows that the use of recycled aggregate in concrete at a proportion of 30% of the natural aggregates (Series 2) improved the thermal conductivity coefficient by approximately 12% compared with the control series. This is mainly due to the reduction in the density of the composite in the presence of recycled aggregate. The use of the additive in the form of a FAS–RCM mixture in an amount of 30% of the cement mass (Series 3) resulted in a reduction in the λ coefficient by about 16% compared with the control series. Thus, the addition of an ash–slag mixture had a positive effect on the thermal parameters of concrete. Although the thermal conductivity coefficient of construction materials is less important than in the case of insulating materials, currently, the aim is to produce construction materials with a low λ value (e.g., ceramic blocks, expanded clay concrete blocks, cellular concrete blocks). Such materials enable the construction of a single-layer wall (without insulation) or one with an insulating layer of the lowest possible thickness.

### 4.4. Microstructural Analysis

#### 4.4.1. Thermogravimetric Analysis and Differential Thermal Analysis (TGA/DTA)

Figure 14 shows the TGA/DTA results for the three mixes (control, FAS–RCM, and FAS–RCM + RCA) after 28 days of curing. The observable mass losses can be divided into five stages as follows. In Stage 1, from ambient temperature to 105 °C, the physically adsorbed water (moisture) was lost. It can be observed that the highest loss was for FAS–RCM + RCA, due to the higher absorption of recycled aggregates compared with natural aggregates. In Stage 2 from 105 to 400 °C, decomposition of aluminates and calcium silicates occurred [46,47,48,49,50]. In Stage 3, from 400 to 460 °C, decomposition of the portlandite occurred [50,51]. From 460 to 640 °C (Stage 4), the mass loss was due to the initial decomposition of carbonates that formed during the hardening process [46]. In the final stage from 640 to 1000 °C, the decomposition of calcium carbonate occurred [46,50].

#### 4.4.2. Scanning Electron Microscopy (SEM) and Optical Microscopy

To observe their microstructure, the samples with the highest compressive strength (FAS–RCM + RCA) and the reference sample (control) were analyzed. Figure 15A,B shows the interfacial transition zone (ITZ) between the NA and cement paste for the control concrete and RCA and cement paste in concrete with (FAS–RCM + RCA).

Si is the main element of natural aggregate (Figure 15A), and the cement paste mainly comprises Ca, among other elements. It can be observed that the ITZ found between cement paste and NA is a classic gap. For the concrete FAS–RCM + RCA (see Figure 15B), the ITZ between the grain of the recycling aggregate and the new cement paste has a compact structure, which may result from the strong connection of high-quality RCA thanks to thermomechanical treatment with the new ingredients of hydration products. These hydration products came from the unhydrated cement found in the RCM and RCA, which was shown by XRD, and improved the compressive strength. A similar result was found by Ismail and Ramli [54].

## 5. Conclusions

Concretes made with FAS, RCM, and RCA were characterized by a higher compressive strength (by about 7%) compared with traditionally produced concrete in the case where 30% of the natural aggregates are replaced by recycled aggregates, with the additional use of the innovative FAS–RCM additive as 30% of the cement mass.The water absorption of concrete with recycled aggregates and the addition of FAS–RCM was very low (not exceeding 3.5%); this value was obtained as a result of the waste FAS–RCM and RCA affecting the sealing of the composite structure. Usually, low water absorption in concrete is obtained as a result of the use of expensive plasticizers or superplasticizers.Concretes based on FAS–RCM and RCA were characterized by low dry and saturated bulk density values (not exceeding 2.5 kg/dm^3^ in the case of 30% recycled aggregates).Concretes with the addition of FAS–RCM and RCA performed better compared with the control in terms of frost resistance. In the tests carried out on concrete containing 100% recycled aggregates, they could be classified into the high F200 class (weight loss < 5%, strength decrease < 20%), while concretes containing 30% RCA and 30% FAS–RCM tested with de-icing salts could be classified as FT1 (mean weight of exfoliation ≤ 1.0, no result > 1.5).Concrete with waste materials had a lower thermal conductivity coefficient than standard composites because of the waste materials (FAS–RCM and RCA).The portlandite and larnite phases (XRD) found in recycled cement mortar and recycled concrete aggregate improved the compressive strengthSEM showed cement hydration products at the interfacial transition zone (ITZ) between the recycled aggregate and the cement paste.

## Figures and Tables

**Figure 1 materials-15-01438-f001:**
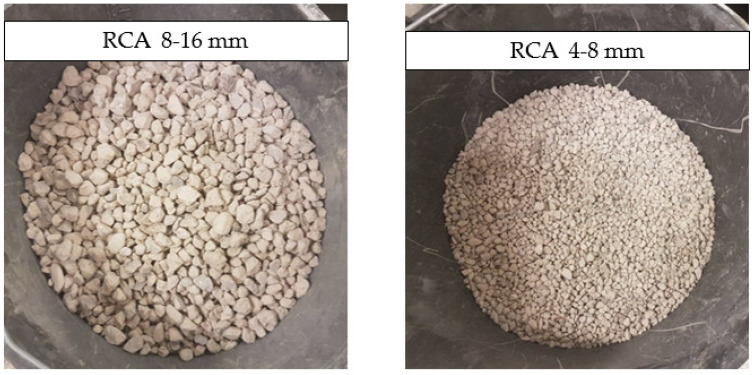
Recycled concrete aggregate (RCA): 8–16 mm fraction (on the **left**) and 4–16 mm fraction (on the **right**).

**Figure 2 materials-15-01438-f002:**
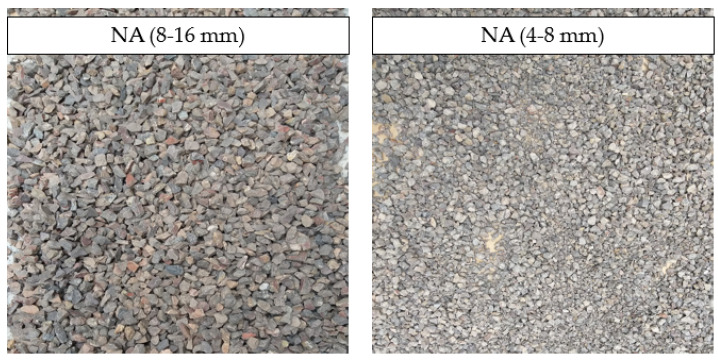
Natural aggregate (NA): dolomite fractions of 8–16 mm (on the **left**) and 4–8 mm (on the **right**).

**Figure 3 materials-15-01438-f003:**
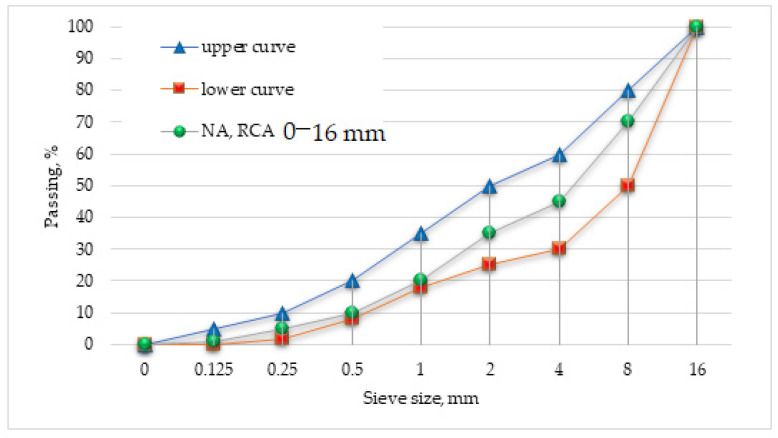
Gradation curves of the NA and RCA.

**Figure 4 materials-15-01438-f004:**
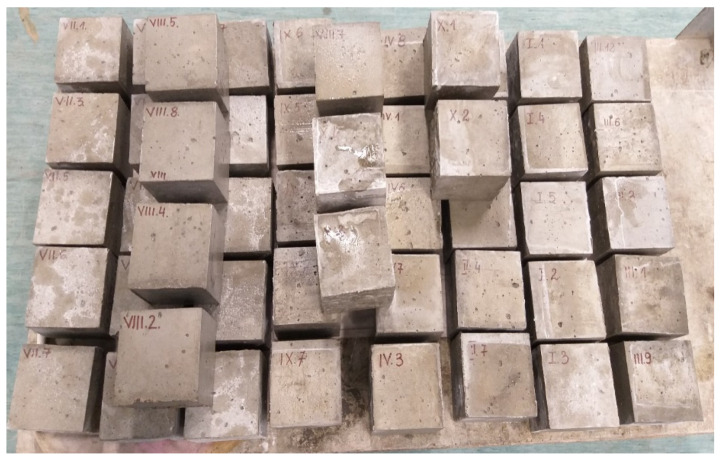
Concrete cubes prepared for testing.

**Figure 5 materials-15-01438-f005:**
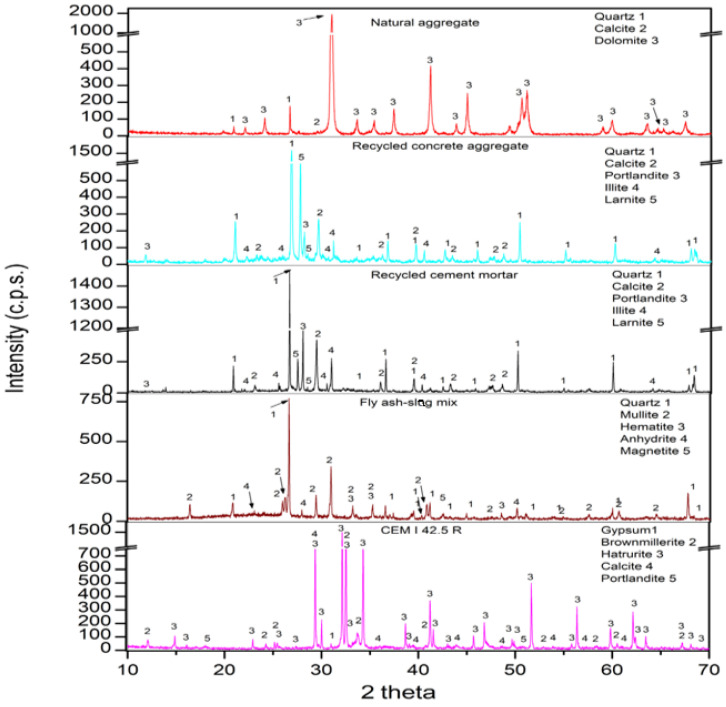
X-ray diffraction patterns of the raw materials.

**Figure 6 materials-15-01438-f006:**
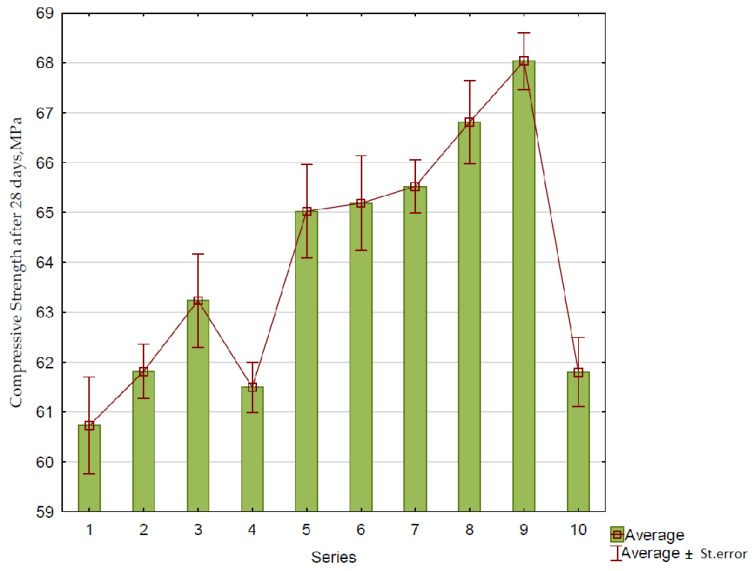
Average results of the compressive strength (MPa) test after 28 days of maturation in comparison with the control series (No. 10).

**Figure 7 materials-15-01438-f007:**
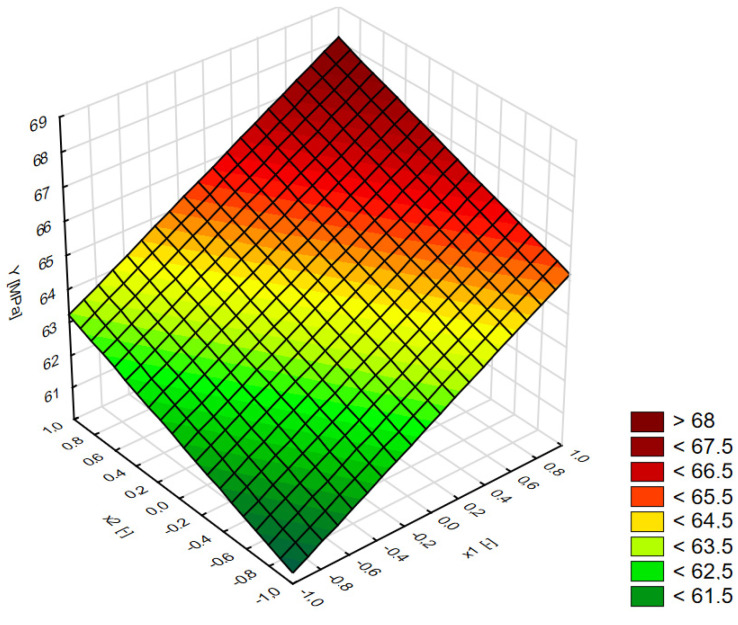
The changes in compressive strength (Y) of concrete depending on the amount of RCM (*x*_1_) and the amount of FAS–RCM (*x*_2_).

**Figure 8 materials-15-01438-f008:**
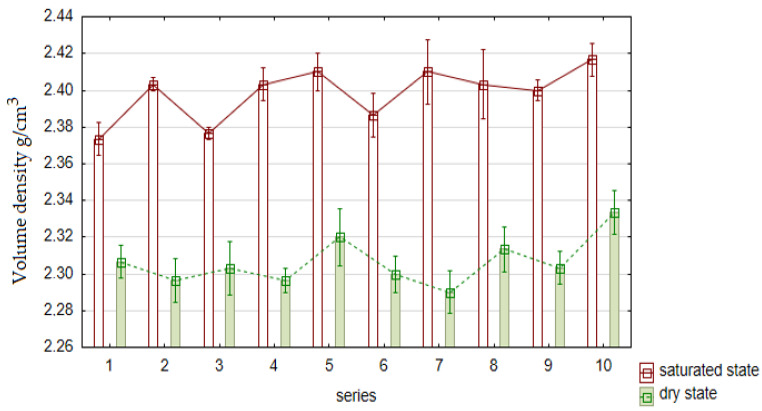
Average results of the volume density of concrete (g/cm^3^) in the saturated and dry states after 28 days of maturation in comparison with the control series (No. 10).

**Figure 9 materials-15-01438-f009:**
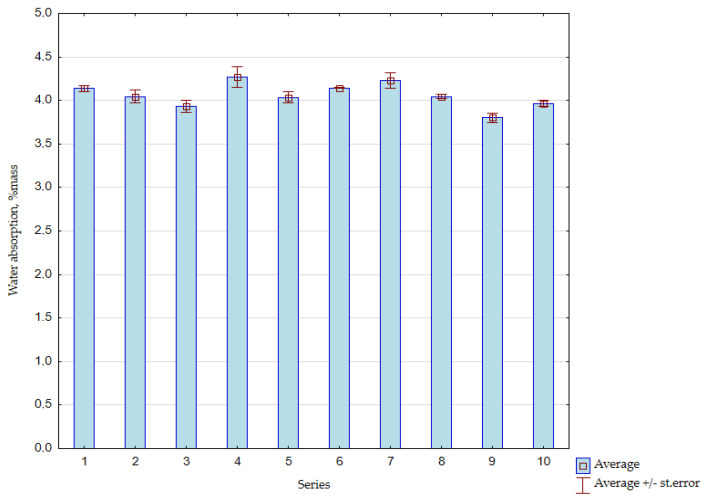
The average results of the water absorption tests for each series of concrete after 28 days, %mass.

**Figure 10 materials-15-01438-f010:**
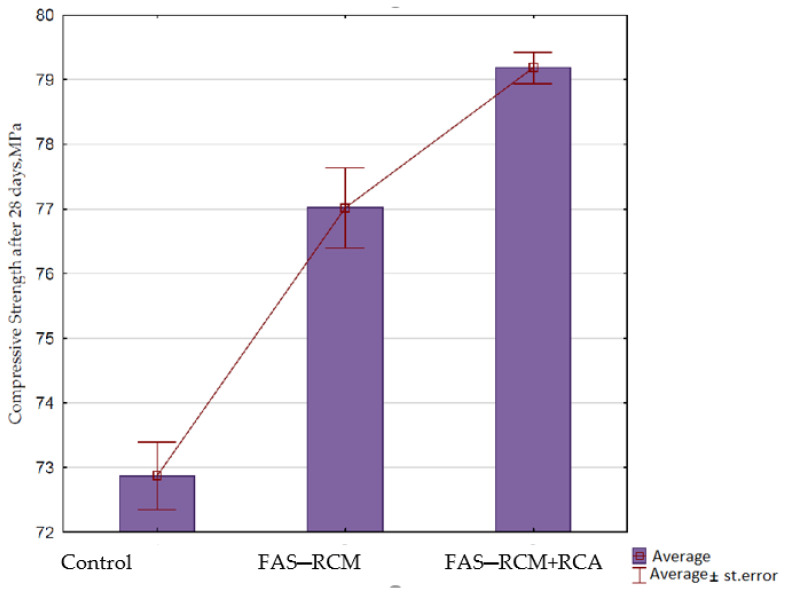
Average results of the compressive strength test of concrete after 28 days of maturation, MPa.

**Figure 11 materials-15-01438-f011:**
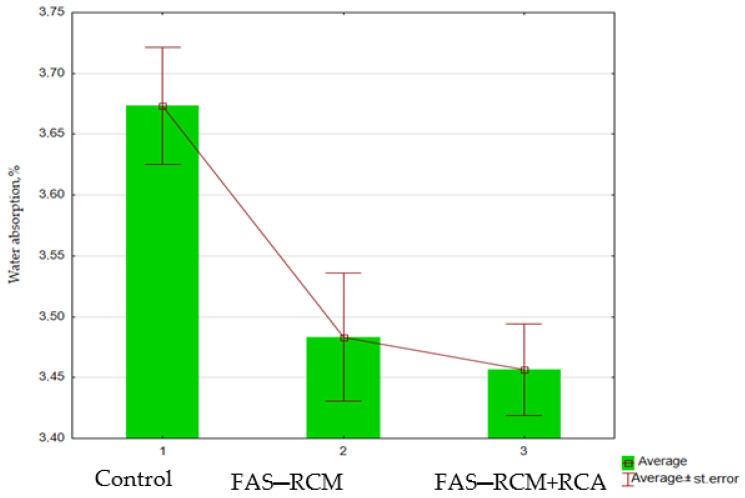
The average water absorption of concrete, %mass.

**Figure 12 materials-15-01438-f012:**
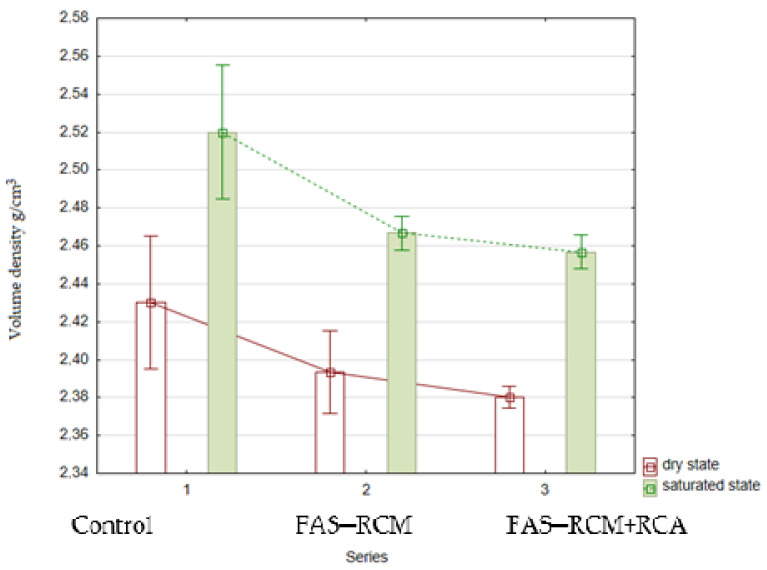
The average volume density of concrete in the saturated and dry state.

**Figure 13 materials-15-01438-f013:**
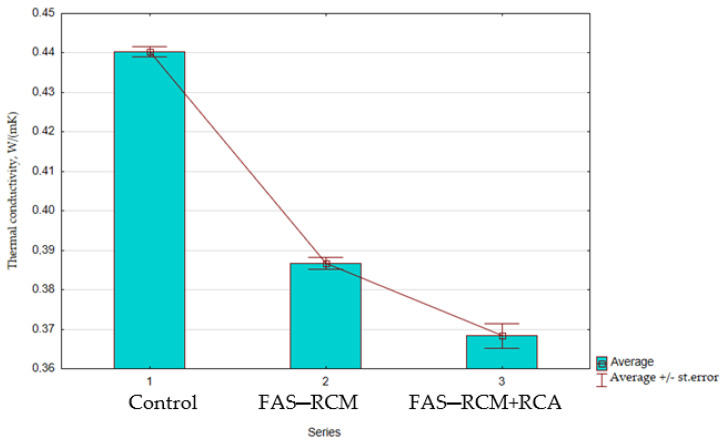
The average thermal conductivity results of concrete, W/(m·K).

**Figure 14 materials-15-01438-f014:**
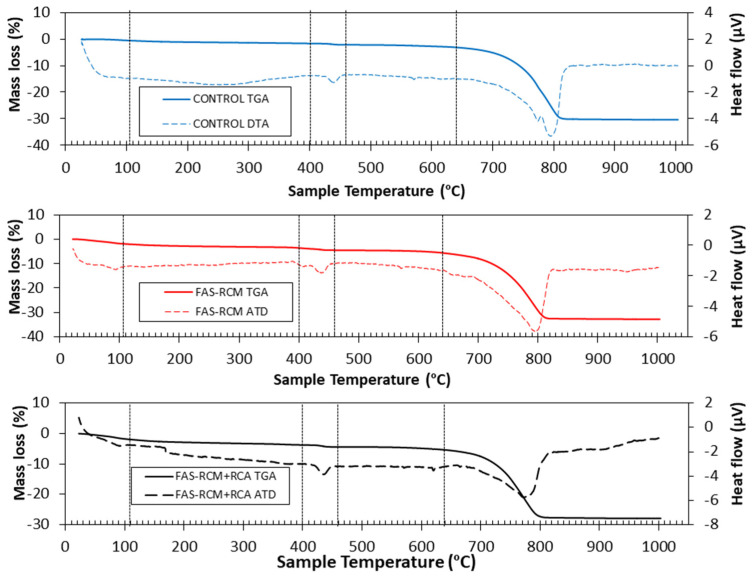
The TGA (solid lines) and DTA (dotted lines) for control, FAS–RCM, and FAS–RCM + RCA concrete after maturing for 28 days.

**Figure 15 materials-15-01438-f015:**
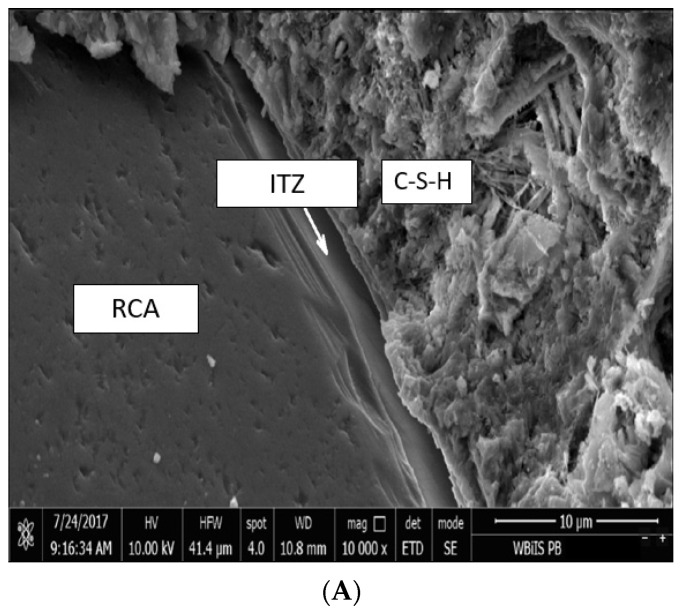
(**A**) SEM images for concrete reference mag. ×10,000. Interfacial transition zone (ITZ) between the NA and cement paste. (**B**) SEM images for concrete with (FAS–RCM + RCA), mag. ×4000.

**Table 1 materials-15-01438-t001:** The properties of the NA and RCA.

Properties	Unit	NA		RCA	
		4–8 mm	8–16 mm	4–8 mm	8–16 mm
Volume density, ρ_a_	g/cm^3^	2.82	2.82	2.75	2.78
Dry volume density, ρ_rd_	g/cm^3^	2.72	2.72	2.44	2.46
Saturated volume density, ρ_ssd_	g/cm^3^	2.76	2.77	2.54	2.51
Water absorption, WA_24_	%	1.5	1.5	2.0	1.7

**Table 2 materials-15-01438-t002:** Sieving results of the recycled cement mortar (RCM) and the fly ash–slag mix (FAS).

Sieve Size, mm	0.063	0.125	0.250	0.50	1.00	2.00	4.00
Passing, % (RCM)	32.4	47.4	61.8	66	83.3	96.9	100
Passing, % (FAS)	42.1	60.4	76.6	81	93.4	98.8	100

**Table 3 materials-15-01438-t003:** Physical properties of CEM I 42.5 R, FAS, and RCM.

Properties	CEM I 42.5R	FAS	RCM
Skeletal density, g/cm^3^	3.05	2.35	2.66
Bulk density, g/cm^3^	1.40	1.22	1.35
Specific surface area according to Blaine’s method, cm^2^/g	4210	2450	1650

**Table 4 materials-15-01438-t004:** Variables in the experimental plan.

*X* _1_	Amount of RCM, % of the summary mass of FAS–RCM	0	25	50
*X* _2_	Amount of FAS–RCM, % of cement mass	10	20	30

**Table 5 materials-15-01438-t005:** The experimental plan.

Series	Variables	
	X_1_, %	X_2_, %
1	0	10
2	0	20
3	0	30
4	25	10
5	25	20
6	25	30
7	50	10
8	50	20
9	50	30
10	0	0

**Table 6 materials-15-01438-t006:** The concrete mixture composition for 1 m^3^.

Series	Amount									
	CEM I 42.5R	Water	SP	*w/s*	RCA	FAS	Sand 0–2 mm	Aggregate 2–4 mm	Aggregate 4–8 mm	Aggregate 8–16 mm
	kg	dm^3^	dm^3^	-	kg	kg	kg	kg	kg	kg
1	345.6				0.0	36.0	657.9	188.0	470.0	564.0
2	331.2				0.0	72.0	646.4	184.7	461.7	554.0
3	316.8				0.0	108.0	634.8	181.4	453.4	544.1
4	345.6				9.0	27.0	658.8	188.2	470.6	564.7
5	331.2	156.6	5.4	0.45	18.0	54.0	648.1	185.2	463.0	555.5
6	316.8				27.0	81.0	637.4	182.1	455.3	546.4
7	345.6				18.0	18.0	659.6	188.5	471.2	565.4
8	331.2				36.0	36.0	649.7	185.6	464.1	556.9
9	316.8				54.0	54.0	639.8	182.8	457.0	548.4
Control Series										
10	360.0	156.6	5.4	0.45	0.0	0.0	669.5	191.3	478.2	573.9

**Table 7 materials-15-01438-t007:** The test plan and concrete mix compositions per 1 m^3^.

Material	Unit	Series 1 (Control)	Series 2 (FAS–RCM)	Series 3 (FAS–RCM + RCA)
Cement CEM I 42,5R	kg	360	360	317
*w/s*	-	0.45	0.45	0.45
Water	dm^3^	156.6	156.6	156.6
SP	dm^3^	5.4	5.4	5.4
FAS		-	-	54.0
RCM		-	-	54.0
Sand 0–2 mm		697.5	691.5	660.9
NA 2–4 mm		199.3	197.6	188.8
NA 4–8 mm	kg	498.2	345.8	330.4
NA 8–16 mm		597.8	414.9	396.5
RCA 4–8 mm		-	148.2	141.6
RCA 8–16 mm		-	177.8	169.9

**Table 8 materials-15-01438-t008:** XRF chemical composition of the raw materials, percentage (%).

Oxides, %	Cement	Natural Aggregate (NA)	Recycled Concrete Aggregate (RCA)	Recycled Cement Mortar (RCM)	Fly Ash–Slag Mix
Na_2_O	0.26	-	0.34	0.37	7.10
MgO	0.55	17.25	2.31	2.54	1.22
Al_2_O_3_	3.34	2.38	5.86	6.44	13.32
SiO_2_	13.93	6.74	24.39	26.96	42.57
P_2_O_5_	0.21	-	0.28	0.31	0.24
SO_3_	4.02	0.07	1.47	1.63	0.20
Cl_2_O_3_	0.18	0.06	0.06	0.07	0.03
K_2_O	0.58	0.92	1.09	1.22	2.12
CaO	54.29	27.28	27.56	30.50	5.05
TiO_2_	0.40	-	0.31	0.34	0.60
MnO_2_	0.07	0.15	-	0.07	0.04
Fe_2_O_3_	2.86	0.98	1.87	2.07	2.69
CuO	0.02	-	-	-	-
ZnO	0.08	-	-	0.01	-
SrO	0.17	-	0.04	0.04	0.04
ZrO_2_	-	-	-	-	-
BaO	-	-	-	-	-
Cr_2_O_3_	-	-	-	0.02	-
Balance CO_2_,%	19.05	44.17	34.34	27.43	24.78
Total,%	100	100	100	100	100

**Table 9 materials-15-01438-t009:** Frost resistance results after 28 cycles of freezing and thawing.

Series	Surface Area of Exfoliation	Mass of Exfoliation	Mass Loss after 28 Cycles, *m28*	Average Mass Loss, *m28*
	m^2^	kg	kg/m^2^	kg/m^2^
3	0.0063	0.00548	0.87	0.91
	0.0064	0.00784	1.23	
	0.0072	0.00457	0.63	
9	0.0072	0.00319	0.44	0.83
	0.0064	0.00462	0.72	
	0.0064	0.00621	0.97	
10	0.0060	0.00474	0.79	0.96
	0.0064	0.00785	1.23	
	0.0067	0.00585	0.87	

## Data Availability

Not applicable.

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
