# Peer review of "The Performance of Concrete Made with Secondary Products—Recycled Coarse Aggregates, Recycled Cement Mortar, and Fly Ash–Slag Mix"

_materials, 2022, doi:10.3390/ma15041438_

Round 1
Reviewer 1 Report
Manuscript ID: materials-1577752 presents an interesting topic, however, the paper needs a major revision to improve the scientific soundness of paper and to enhance the attraction to readers of the journal. The study is current and comprehensive but needs to be presented in a very academic way of article writing.
- On line 42, the concrete industry is one of the most environmentally damaging in the world, mainly due to the consumption of water and aggregates and the production of cement-Please revise the statement as it’s the first line of the introduction section. The first line of the introduction should not be contradictory. Or, write it another way.
- On line 59, the “worldwide” word is probably unnecessary here.
- On line 69, Japan is the leader in the utilization of coal combustion by-products (utilization rate above 90%), please provide a reference.
- The photos of the Figures 4 & 5 do not make any sense, please replace them.
- What is the significance of the XRD test results in the study? Please mention this in the XRD section. Why did you conduct this test and what value does the test add to the paper? Please note that the XRD test of cement, sand, natural aggregates were conducted by many researchers before you.
- Section 4.4.2 has 4 images, one image (on the top right) needs numbering and explanation of what authors want to show readers. Every photograph should have individual numbering. Also, this section needs to be improved as all statements are very general, authors should critically think on it, especially, cause and effects should be described.
- Concrete with the addition of FAS-RCM and RCA are resistant to frost, please revise as the statement is not acceptable. You can write it performs better compared to the control.
- Poor academic writing, please revise the whole paper with the help of someone expert in academic writing and data presentation.
- Many sections just present the data of the study, please analyse the data critically to present your findings and comment on the cause and effect.
- Please revise the title, many researchers already published papers on the topic. So, you can’t write a proposal in the title. Also, you can describe that concrete made from recycled coarse aggregates, recycled cement is eco-friendly in the introduction section. Please remove the eco-friendly word from the title. One title of the paper could be “performance of concrete made from/of recycled coarse aggregates, recycled cement mortar and Fly Ash-Slag”.
Author Response
We would like to thank you for your review and your time spent to improve our article. We hope that the changes made will make the article available for publication. On behalf of all authors, Katarzyna Kalinowska-Wichrowska

Reviewer 2 Report
Lines 130-132, the cement mortar obtained during Los Angeles drum mechanical treatment is actually the fine residues containing cement and sand particles. Therefore, authors need to replace the term "recycled cement mortar (RCM)" with "recycled fine aggregates" or any other suitable term throughout the manuscript.
The introduction section needs a lot of improvement in the form of more details of past relevant studies and the inclusion of the latest studies related to recycled aggregate concrete and the inferior properties of recycled concrete aggregates. Following study should be helpful: "Axial stress-strain performance of steel spiral confined acetic acid immersed and mechanically rubbed recycled aggregate concrete".
Please provide the list of abbreviations.
Authors need to add the significance section of this study with reference to past studies for highlighting the novelty of this study. What is the novelty of this work? please discuss with reference to similar past studies or reports.
Authors need to compare the results obtained in this study with similar previous studies and provide justification for the result trends with references.
The authors should add a cost to benefit analysis section in this study. The treatment of fly ash and recycled coarse aggregates performed in this study is very energy-intensive and costly. Please justify the cost of treatment. Cost comparison of recycled aggregate concrete and traditional concrete will be a very important discussion in this study.
Author Response

(The authors gave the same response as above.)

Reviewer 3 Report
The article entitled "The Proposal of Eco-friendly Concrete with Secondary Products - Recycled Coarse Aggregates, Recycled Cement Mortar and Fly Ash-Slag Mix" fits into the thematic focus of the Materials magazine, section "Construction and Building Materials".
The article has the structure of a research article. The literature is fully related to the issue of the article. All literary references are cited in the article. After studying the article, I agree with the authors' conclusions. Nevertheless, I have a few comments that need to be incorporated.
Comments:
1) line 62 - correct literature citations on [13-17]
2) Unify units in tab. 1 and 3 …density…… per g/cm3
3) line 334 - correct unit on (g = 9.81 m / s2)
4) line 360 - In Table 8 top up the unit. In my opinion, this is the proportion of oxides in% (total 100)
5) line 391 - Add error bars in Figure 8.
6) Were initial strength tests also performed? E.g. after 2, 7 days.
7) Have long-term strength tests been performed? E.g. after 90, 180 days.
8) line 419 – Inf Figure 10 correct the superscript of the unit on the Y-axis.
9) line 463 - Add error bars in Figure 12.
10) line 488 - in Figure 13 remove % for numbers on the Y axis and for column values in the graph
Author Response

(The authors gave the same response as above.)

Reviewer 4 Report
The research article "Products - Recycled Coarse Aggregates, Recycled Cement 3 Mortar and Fly Ash-Slag Mix" is comprehensive and well written.
- There is a need to give more details and reasons for the selected percentages of the replacement. For example 10-30% or 0-50% on what basis and how it could justify the cost and other aspects.
- On the other hand, increasing the content of recycled mortar in the FAS-RCM mass from 0 to 50% resulted in a sig-406 significant increase in compressive strength by an average of 8–14%. The authors reported significant, whereas its only maximum of 14%, which is not much significant from practical points of view, there is need to justify.
- Similarly in experimental program 2, the % increase is very small.
- The authors are suggested to include figures of tested samples under compressions and include some discussion on failure modes.
- It's better to provide some scientific reasons for the observed mechanisms, for example, The portlandite and larnite phases (XRD) found in recycled cement mortar and recycled concrete aggregate improved the compressive strength. Usually, the recycled concrete aggregates are not helpful to improve the compressive strength.
Author Response

(The authors gave the same response as above.)

Round 2
Reviewer 1 Report
- The authors have improved the quality of the paper and addressed my questions and comments except comment number 5 regarding XRD.
- Therefore, I again suggest the authors to improve the XRD section.
Author Response
Dear Reviewer,
We would like to thank you for your review. We hope that the changes made will make the article available for publication.
We have corrected the paragraph on research XRD (we marked on red and yellow).
About English revision- we used MDPI English Revision Office and I am enclosing the certificate confirming it.
On behalf of all authors, Katarzyna Kalinowska-Wichrowska

Reviewer 2 Report
The manuscript can be accepted for publication.
Author Response
Dear Reviewer,
We would like to thank you for your review and accept our article for publication.
The English revision has been done by MDPI Editorial Office and I am enclosing the certificate of that.
Thank you.
On behalf of all authors, Katarzyna Kalinowska-Wichrowska
